# RODE: Learning Roles to Decompose Multi-Agent Tasks

**Tonghan Wang**[†]**, Tarun Gupta**[‡]**, Anuj Mahajan**[‡]**, Bei Peng**[‡]
[†] Institute for Interdisciplinary Information Sciences, Tsinghua University
[‡] Univeristy of Oxford
`tonghanwang1996@gmail.com`
`{tarun.gupta, anuj.mahajan, bei.peng}@cs.ox.ac.uk`

**Shimon Whiteson**[*]
Univeristy of Oxford
`shimon.whiteson@cs.ox.ac.uk`

**Chongjie Zhang**[*]
Tsinghua University
`chongjie@tsinghua.edu.cn`

## Abstract

Role-based learning holds the promise of achieving scalable multi-agent learning by decomposing complex tasks using roles. However, it is largely unclear how to efficiently discover such a set of roles. To solve this problem, we propose to first decompose joint action spaces into restricted role action spaces by clustering actions according to their effects on the environment and other agents. Learning a role selector based on action effects makes role discovery much easier because it forms a bi-level learning hierarchy: the role selector searches in a smaller role space and at a lower temporal resolution, while role policies learn in significantly reduced primitive action-observation spaces. We further integrate information about action effects into the role policies to boost learning efficiency and policy generalization. By virtue of these advances, our method (1) outperforms the current state-of-the-art MARL algorithms on 9 of the 14 scenarios that comprise the challenging StarCraft II micromanagement benchmark and (2) achieves rapid transfer to new environments with three times the number of agents. Demonstrative videos can be viewed at *https://sites.google.com/view/rode-marl*.

## 1 Introduction

Cooperative multi-agent problems are ubiquitous in real-world applications, such as crewless aerial vehicles (Pham et al., 2018; Xu et al., 2018) and sensor networks (Zhang & Lesser, 2013). However, learning control policies for such systems remains a major challenge. Joint action learning (Claus & Boutilier, 1998) learns centralized policies conditioned on the full state, but this global information is often unavailable during execution due to partial observability or communication constraints. Independent learning (Tan, 1993) avoids this problem by learning decentralized policies but suffers from non-stationarity during learning as it treats other learning agents as part of the environment.

The framework of centralized training with decentralized execution (CTDE) (Foerster et al., 2016; Gupta et al., 2017; Rashid et al., 2018) combines the advantages of these two paradigms. Decentralized policies are learned in a centralized manner so that they can share information, parameters, etc., without restriction during training. Although CTDE algorithms can solve many multi-agent problems (Mahajan et al., 2019; Das et al., 2019; Wang et al., 2020d), during training they must search in the joint action-observation space, which grows exponentially with the number of agents. This makes it difficult to learn efficiently when the number of agents is large (Samvelyan et al., 2019).

Humans cooperate in a more effective way. When dealing with complex tasks, instead of directly conducting a collective search in the full action-observation space, they typically decompose the task and let sub-groups of individuals learn to solve different sub-tasks (Smith, 1937; Butler, 2012). Once the task is decomposed, the complexity of cooperative learning can be effectively reduced

---

[*]Equal advising

because individuals can focus on restricted sub-problems, each of which often involves a smaller action-observation space. Such potential scalability motivates the use of roles in multi-agent tasks, in which each role is associated with a certain sub-task and a corresponding policy.

The key question in realizing such scalable learning is how to come up with a set of roles to effectively decompose the task. Previous work typically predefines the task decomposition and roles (Pavón & Gómez-Sanz, 2003; Cossentino et al., 2005; Spanoudakis & Moraitis, 2010; Bonjean et al., 2014). However, this requires prior knowledge that might not be available in practice and may prevent the learning methods from transferring to different environments.

Therefore, to be practical, it is crucial for role-based methods to automatically learn an appropriate set of roles. However, learning roles from scratch might not be easier than learning without roles, as directly finding an optimal decomposition suffers from the same problem as other CTDE learning methods – searching in the large joint space with substantial exploration (Wang et al., 2020c).

To solve this problem, we propose a novel framework for learning ROles to DEcompose (RODE) multi-agent tasks. Our key insight is that, instead of learning roles from scratch, role discovery is easier if we first decompose joint action spaces according to action functionality. Intuitively, when cooperating with other agents, only a subset of actions that can fulfill a certain functionality is needed under certain observations. For example, in football games, the player who does not possess the ball only needs to explore how to `move` or `sprint` when attacking. In practice, we propose to first learn effect-based action representations and cluster actions into role action spaces according to their effects on the environment and other agents. Then, with knowledge of effects of available actions, we train a role selector that determines corresponding role observation spaces.

This design forms a bi-level learning framework. At the top level, a role selector coordinates role assignments in a smaller role space and at a lower temporal resolution. At the low level, role policies explore strategies in reduced primitive action-observation spaces. In this way, the learning complexity is significantly reduced by decomposing a multi-agent cooperation problem, both temporally and spatially, into several short-horizon learning problems with fewer agents. To further improve learning efficiency on the sub-problems, we condition role policies on the learned effect-based action representations, which improves generalizability of role policies across actions.

We test RODE on StarCraft II micromanagement environments (Samvelyan et al., 2019). Results on this benchmark show that RODE establishes a new state of the art. Particularly, RODE has the best performance on 9 out of all 14 maps, including all 5 super hard maps and most hard maps. Visualizations of learned action representations, factored action spaces, and dynamics of role selections shed further light on the superior performance of RODE. We also demonstrate that conditioning the role selector and role policies on action representations enables learned RODE policies to be transferred to tasks with different numbers of actions and agents, including tasks with three times as many agents.

## 2 RODE Learning Framework

In this section, we introduce the RODE learning framework. We consider fully cooperative multi-agent tasks that can be modelled as a Dec-POMDP (Oliehoek et al., 2016) consisting of a tuple $G=\langle I, S, A, P, R, \Omega, O, n, \gamma\rangle$, where $I$ is the finite set of $n$ agents, $\gamma \in [0, 1)$ is the discount factor, and $s \in S$ is the true state of the environment. At each timestep, each agent $i$ receives an observation $o_i \in \Omega$ drawn according to the observation function $O(s, i)$ and selects an action $a_i \in A$, forming a joint action $\boldsymbol{a} \in A^n$, leading to a next state $s'$ according to the transition function $P(s'|s, \boldsymbol{a})$, and observing a reward $r = R(s, \boldsymbol{a})$ shared by all agents. Each agent has local action-observation history $\tau_i \in \mathrm{T} \equiv (\Omega \times A)^*$.

Our idea is to learn to decompose a multi-agent cooperative task into a set of sub-tasks, each of which has a much smaller action-observation space. Each sub-task is associated with a role, and agents taking the same role collectively learn a role policy for solving the sub-task by sharing their learning. Formally, we propose the following definition of sub-tasks and roles.

**Definition 1** (Role and Sub-Task). *Given a cooperative multi-agent task $G=\langle I, S, A, P, R, \Omega, O, n, \gamma\rangle$, let $\Psi$ be a set of roles. A role $\rho_j \in \Psi$ is a tuple $\langle g_j, \pi_{\rho_j}\rangle$ where $g_j = \langle I_j, S, A_j, P, R, \Omega_j, O, \gamma\rangle$ is a sub-task and $I_j \subset I$, $\cup_j I_j = I$, and $I_j \cap I_k = \varnothing$, $j \neq k$. $A_j$ is the action space of role $j$,*

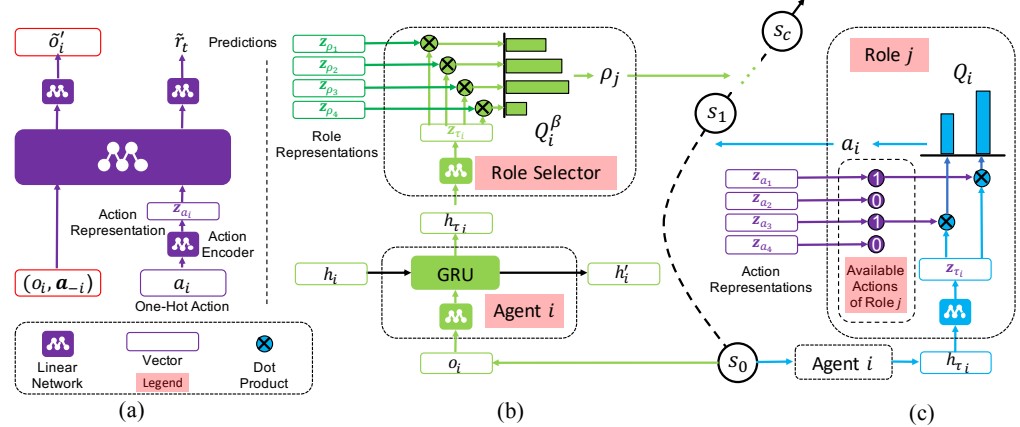

Figure 1: RODE framework. (a) The forward model for learning action representations. (b) Role selector architecture. (c) Role action spaces and role policy structure.

$A_j \subset A$, $\cup_j A_j = A$, and we allow action spaces of different roles to overlap: $|A_j \cap A_k| \geq 0$, $j \neq k$. $\pi_{\rho_j} : \mathrm{T} \times A_j \to [0,1]$ is a role policy for the sub-task $g_j$ associated with the role.

In our definition, a given set of roles specifies not only a factorization of the task but also policies for sub-tasks. This role-based formulation is effectively a kind of dynamic grouping mechanism where agents are grouped by roles. The benefit is that we can learn policies for roles instead of agents, providing scalability with the number of agents and reducing search space of target policies. In our framework, an agent decides which role to take by a *role selector* $\beta$ that is shared among agents. Here, $\beta : \mathrm{T} \to \Psi$ is a deterministic function that is conditioned on local action-observation history.

Our aim is to learn a set of roles $\Psi^*$ that maximizes the expected global return $Q^{\Psi}(s_t, \boldsymbol{a}_t) = \mathbb{E}_{s_{t+1:\infty}, \boldsymbol{a}_{t+1:\infty}}[\sum_{i=0}^{\infty} \gamma^i r_{t+i} | s_t, \boldsymbol{a}_t, \Psi]$. Learning roles involves two questions: learning sub-tasks and role policies. For learning sub-tasks, we need to learn the role selector, $\beta$, and to determine role action spaces, $A_j$. These two components determine the observation and action spaces of roles, respectively. We now introduce the framework to learn these components, which is shown in Fig. 1.

## 2.1 DETERMINING ROLE ACTION SPACES BY LEARNING ACTION REPRESENTATIONS

Restricting the action space of each role can significantly reduce the policy search space. The key to our approach is to factor the action space according to actions' properties and thereby let roles focus on actions with similar effects, such as attacking enemy units of the same type in StarCraft II.

To this end, we learn action representations that can reflect the effects of actions on the environment and other agents. The effect of an action can be measured by the induced reward and the change in local observations. Therefore, we formulate an objective for learning action representations that incentivizes including enough information such that the next observations and rewards can be predicted when given the actions of other agents and current observations.

In practice, we use the predictive model shown in Fig. 1(a) to learn an action encoder $f_e(\cdot; \theta_e)$: $\mathbb{R}^{|A|} \to \mathbb{R}^d$. This encoder, parameterized by $\theta_e$, maps one-hot actions to a $d$-dimensional representation space. We denote the representation of an action $a$ in this space by $\boldsymbol{z}_a$, *i.e.*, $\boldsymbol{z}_a = f_e(a; \theta_e)$, which is then used to predict the next observation $o'_i$ and the global reward $r$, given the current observation $o_i$ of an agent $i$, and the one-hot actions of other agents, $\boldsymbol{a}_{-i}$. This model can be interpreted as a forward model, which is trained by minimizing the following loss function:

$$\mathcal{L}_e(\theta_e, \xi_e) = \mathbb{E}_{(\boldsymbol{o}, \boldsymbol{a}, r, \boldsymbol{o}') \sim \mathcal{D}} \left[ \sum_i \|p_o(\boldsymbol{z}_{a_i}, o_i, \boldsymbol{a}_{-i}) - o'_i\|_2^2 + \lambda_e \sum_i (p_r(\boldsymbol{z}_{a_i}, o_i, \boldsymbol{a}_{-i}) - r)^2 \right], \quad (1)$$

where $p_o$ and $p_r$ are predictors for observations and rewards, respectively, and parameterized by $\xi_e$. $\lambda_e$ is a scaling factor, $\mathcal{D}$ is an replay buffer, and the sum is carried out over all the agents.

In the beginning, we initialize $K$ roles, each with the full action space. After collecting samples and training the predictive model for $t_e$ timesteps, we cluster the actions according to their latent representations and set the action space of each role to contain one of the clusters. Outliers are added to all the clusters to avoid action sets consisting of only one action. After this update, training begins and action representations and action spaces of each role are kept fixed during training.

In this paper, we simply use $k$-means clustering based on Euclidean distances between action representations, but it can be easily extended to other clustering methods. In Appendix E, we investigate the influence of different $k$ values and how to determine the number of clusters automatically. In practice, we find that the forward model converges quickly. For example, a well-formed representation space can be learned using $50K$ samples on StarCraft II micromanagement tasks, while the training usually lasts for $2M$ timesteps.

## 2.2 Learning the Role Selector and Role Policies

Learning action representations to cluster actions based on their effects leads to a factorization of the action space, where each subset of actions can fulfill a certain functionality. To make full use of such a factorization, we use a bi-level hierarchical structure to coordinate the selection of roles and primitive actions. At the top level, a role selector assigns a role to each agent every $c$ timesteps. After a role is assigned, an agent explores in the corresponding restricted role action space to learn the role policy. We now describe how to design efficient, transferable, and lightweight structures for the role selector and role policies.

For the role selector, we can simply use a conventional $Q$-network whose input is local action-observation history and output is $Q$-values for each role. However, this structure may not be efficient because it ignores information about action space of different roles. Intuitively, selecting a role is selecting a subset of actions to execute for the next $c$ timesteps, so the $Q$-value of selecting a role is closely related to its role action space. Therefore, we propose to base the role selector on the average representation of available actions. Fig. 1(b) shows the proposed role selector structure. Specifically, we call

$$\boldsymbol{z}_{\rho_j} = \frac{1}{|A_j|} \sum_{a_k \in A_j} \boldsymbol{z}_{a_k}, \tag{2}$$

the representation of role $\rho_j$, where $A_j$ is its restricted action space. To select roles, agents share a linear layer and a GRU, parameterized by $\theta_{\tau_\beta}$, to encode the local action-observation history $\tau$ into a fixed-length vector $h_\tau$. The role selector, a fully connected network $f_\beta(h_\tau; \theta_\beta)$ with parameters $\theta_\beta$, then maps $h_\tau$ to $\boldsymbol{z}_\tau \in \mathbb{R}^d$, and we estimate the expected return of agent $i$ selecting role $\rho_j$ as:

$$Q_i^\beta(\tau_i, \rho_j) = \boldsymbol{z}_{\tau_i}^{\mathrm{T}} \boldsymbol{z}_{\rho_j}. \tag{3}$$

Agents select their roles concurrently, which may lead to unintended behaviors. For example, all agents choosing to attack a certain unit type in StarCraft II causes overkill. To better coordinate role assignments, we use a mixing network to estimate global $Q$-values, $Q_{tot}^\beta$, using per-agent utility $Q_i^\beta$ so that the role selector can be trained using global rewards. In this paper, we use the mixing network introduced by QMIX (Rashid et al., 2018) for its monotonic approximation. The parameters of the mixing network are conditioned on the global state $s$ and are generated by a hypernetwork (Ha et al., 2016) parameterized by $\xi_\beta$. Then we minimize the following TD loss to update the role selector $f_\beta$:

$$\mathcal{L}_\beta(\theta_{\tau_\beta}, \theta_\beta, \xi_\beta) = \mathbb{E}_\mathcal{D} \left[ \left( \sum_{t'=0}^{c-1} r_{t+t'} + \gamma \max_{\boldsymbol{\rho}'} \bar{Q}_{tot}^\beta(s_{t+c}, \boldsymbol{\rho}') - Q_{tot}^\beta(s_t, \boldsymbol{\rho}_t) \right)^2 \right], \tag{4}$$

where $\bar{Q}_{tot}^\beta$ is a target network, $\boldsymbol{\rho} = \langle \rho_1, \rho_2, \ldots, \rho_n \rangle$ is the joint role of all agents, and the expectation is estimated with uniform samples from a replay buffer $\mathcal{D}$.

After a role is assigned, an agent follows it for the next $c$ timesteps, during which it can only take actions in the corresponding role action space. Each role $\rho_j$ has a role policy $\pi_{\rho_j} : \mathrm{T} \times A_j \to [0, 1]$ that is defined in the restricted action space. Similar to the role selector, the most straightforward approach to learning role policies is with a conventional deep $Q$-network that directly estimates the $Q$-values of each action. However, basing $Q$-values on action representations makes full use of the effect information of actions and can generalize better across actions.

Therefore, we use the framework shown in Fig. 1(c) for learning role policies. Specifically, agents again use a shared linear layer and a GRU which together encode a local action-observation history $\tau$ into a vector $h_\tau$. We denote the parameters of these two networks by $\theta_{\tau_\rho}$. Each role policy is a fully connected network $f_{\rho_j}(h_\tau; \theta_{\rho_j})$ with parameters $\theta_{\rho_j}$, whose input is $h_\tau$. $f_{\rho_j}$ maps $h_\tau$ to $z_\tau$, which is a vector in $\mathbb{R}^d$. Using action representations $z_{a_k}$ and $z_\tau$, we estimate the value of agent $i$ choosing a primitive action $a_k$ as:

$$Q_i(\tau_i, a_k) = z_{\tau_i}^{\mathrm{T}} z_{a_k}. \tag{5}$$

For learning the $Q_i$ using global rewards, we again feed local $Q$-values into a QMIX-style mixing network to estimate the global action-value, $Q_{tot}(s, \boldsymbol{a})$. The parameters of the mixing network are denoted by $\xi_\rho$. This formulation gives the TD loss for learning role policies:

$$\mathcal{L}_\rho(\theta_{\tau_\rho}, \theta_\rho, \xi_\rho) = \mathbb{E}_{\mathcal{D}}\left[\left(r + \gamma \max_{\boldsymbol{a}'} \bar{Q}_{tot}(s', \boldsymbol{a}') - Q_{tot}(s, \boldsymbol{a})\right)^2\right]. \tag{6}$$

Here, $\bar{Q}_{tot}$ is a target network, $\theta_\rho$ is the parameters of all role policies, and the expectation is estimated with uniform samples from the same replay buffer $\mathcal{D}$ as the one for the role selector. Moreover, two mixing networks for training the role selector and role policies are only used during training. By virtue of effect-based action representations, our framework is lightweight – in practice, we use a single linear layer without activation functions for role policies and a two-layer fully-connected network for the role selector. Basing the role selector and role policies on action representations also enable RODE policies to be rapidly transferred to new tasks with different numbers of agents and actions, which we discuss in detail in Sec. 4.3.

## 3 RELATED WORK

**Hierarchical MARL** To realize efficient role-based learning, RODE adopts a hierarchical decision-making structure. Hierarchical reinforcement learning (HRL) (Sutton et al., 1999; Al-Emran, 2015) has been extensively studied to address the sparse reward problem and to facilitate transfer learning. Single-agent HRL focuses on learning temporal decomposition of tasks, either by learning sub-goals (Nachum et al., 2018b;a; Levy et al., 2018; Ghosh et al., 2018; Sukhbaatar et al., 2018; Dwiel et al., 2019; Nair & Finn, 2019; Nasiriany et al., 2019; Dilokthanakul et al., 2019) or by discovering reusable skills (Daniel et al., 2012; Gregor et al., 2016; Warde-Farley et al., 2018; Shankar & Gupta, 2020; Thomas et al., 2018; Sharma et al., 2020). In multi-agent settings, many challenges, such as efficient communication (Ossenkopf et al., 2019) and labor division (Wang et al., 2020c) in large systems, necessitate hierarchical learning along the second dimension – over agents (Zhang et al., 2010). Ahilan & Dayan (2019) propose a hierarchy in FeUdal (Dayan & Hinton, 1993; Vezhnevets et al., 2017) style for cooperative multi-agent tasks, where managers and workers are predefined and workers are expected to achieve the goal generated by managers. Similar hierarchical organizations among agents have been exploited to control traffic lights (Jin & Ma, 2018). Lee et al. (2019); Yang et al. (2019) propose to use bi-level hierarchies to train (or discover) and coordinate individual skills. Vezhnevets et al. (2020) extend hierarchical MARL to Markov Games, where a high-level policy chooses strategic responses to opponents. RODE proposes to learn multi-agent coordination skills in restricted action spaces and is thus complementary to these previous works.

**Dealing with Large Discrete Action Spaces** In single-agent settings, being able to reason with a large number of discrete actions is essential to bringing reinforcement learning to a larger class of problems, such as natural language processing (He et al., 2016) and recommender systems (Ayush et al., 2020). Sallans & Hinton (2004); Pazis & Parr (2011) represent actions using binary code to reduce the size of large action spaces. Cui & Khardon (2016; 2018) address scalability issues of planning problems by conducting a gradient-based search on a symbolic representation of the state-action value function. Sharma et al. (2017) show that factoring actions into their primary categories can improve performance of deep RL algorithms on Atari 2600 games (Bellemare et al., 2013). All aforementioned methods assume that a handcrafted binary decomposition of raw actions is provided. Another interesting way to deal with large discrete state-action spaces is by estimating equivalence classes over reward trajectories and learn on quotient MDPs (Mahajan & Tulabandhula, 2017a;b).

Yet another line of research uses continuous action representations to discover underlying structures of large discrete action spaces, thereby easing training. Van Hasselt & Wiering (2009) use policy gradients with continuous actions and select the nearest discrete action. Dulac-Arnold et al. (2015)

Move in four directions    Stop    Attack different units    *Noop*      Move    Stop    Attack Zealots    Attack Stalkers    *Noop*

Figure 2: Action representations learned by our predictive model on `corridor` (left) and `3s5z_vs_3s6z` (right). Blue arrows indicate the directions moving towards enemies while red ones show the directions moving away from enemies.

extend this work to large-scale problems but use predefined action embeddings. Tennenholtz & Mannor (2019); Chandak et al. (2019) avoid pre-definitions by extracting action representations from expert demonstrations or without prior knowledge. Action representations have been used to solve natural language processing problems (He et al., 2016), enable generalization to unseen (Ayush et al., 2020) or under-explored (Chandak et al., 2019) actions in RL, learn linear dynamics that help exploration (Kim et al., 2019), and transfer policy across different tasks (Chen et al., 2019). By contrast, we use action representations to factor multi-agent tasks.

Besides learning action representations, Czarnecki et al. (2018); Murali et al. (2018); Farquhar et al. (2020) propose to use curriculum learning over action spaces of increasing sizes to scale learning to large spaces. Although the large action space problem is more demanding in multi-agent settings, how to organize effective learning in them is largely unstudied – most algorithms directly search in the full action spaces, including previous works (Wang et al., 2020f) that consider the effects (semantics) of actions like RODE. This is a key motivation for RODE, where search spaces are reduced by an effect-based action space factorization and a role-based hierarchical structure designed for efficient learning in these factored action spaces.

In Appendix G, we discuss how RODE is related to other role-based learning approaches and multi-agent reinforcement learning algorithms in detail.

## 4 EXPERIMENTS

We design experiments to answer the following questions: (1) Can the proposed predictive model learn action representations that reflect effects of actions? (Sec. 4.1) (2) Can RODE improve learning efficiency? If so, which component contributes the most to the performance gains? (Sec. 4.2) (3) Can RODE support rapid transfer to environments with different numbers of actions and agents? (Sec. 4.3) (4) Can the role selector learn interpretable high-level cooperative behaviors? (Appx. B)

We choose the StarCraft II micromanagement (SMAC) benchmark (Samvelyan et al., 2019) as the testbed for its rich environments and high complexity of control. The SMAC benchmark requires learning policies in a large action space. Agents can `move` in four cardinal directions, `stop`, take `noop` (do nothing), or select an enemy to `attack` at each timestep. Therefore, if there are $n_e$ enemies in the map, the action space for each ally unit consists of $n_e + 6$ discrete actions.

### 4.1 ACTION REPRESENTATIONS

For learning action representations, on all maps, we set the scaling factor $\lambda_e$ to 10 and collect samples and train the predictive model for $50K$ timesteps using the loss function described by Eq. 1. The model is trained every episode using a batch of 32 episodes. In Fig. 2, we show the learned action representations on two maps, `3s5z_vs_3s6z` and `corridor`.

The map `corridor` features homogeneous agents and enemies, with 6 Zealots faced with 24 enemy Zerglings. In this task, all attack actions have similar effects because the enemies are homogeneous. Moreover, due to the symmetry of the map, moving northward and eastward exert similar influence on the environment – moving the agent towards enemies. Likewise, moving southward and westward have the same effect because both actions move agents away from enemies. The learned action encoder captures these underlying structures in the action space (Fig. 2 left). We can observe three

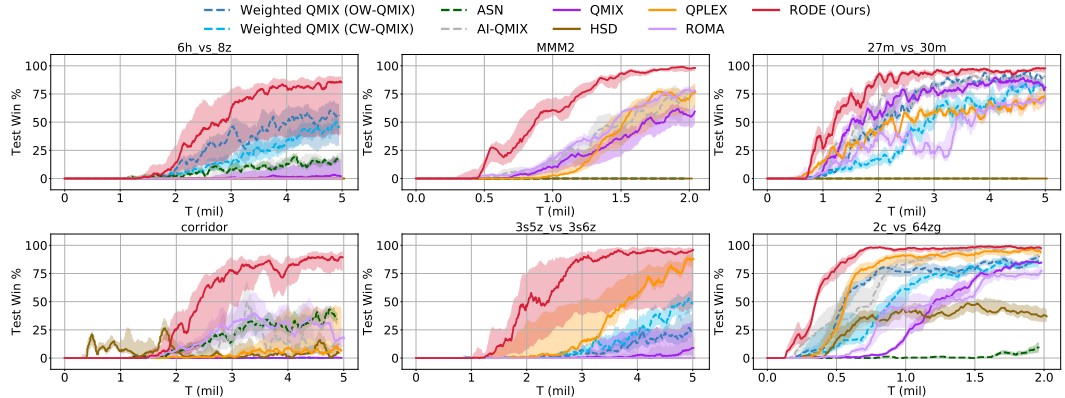

Figure 3: Performance comparison with baselines on *all* **super hard** maps and one **hard** map (2c_vs_64zg). In Appendix C.1, we show results on the whole benchmark.

clear clusters in the latent action representation space, which correspond to the aforementioned three types of actions, respectively. On the map 3s5z_vs_3s6z, 3 Stalkers and 5 Zealots try to beat 3 enemy Stalkers and 6 Enemy Zealots. Unlike corridor, the enemy team consists of heterogeneous units, and attacking Stalkers and Zealots have distinct effects. Our predictive model highlights this difference – the attack actions are clustered into two groups (Fig. 2 right). Moreover, the learned action encoder also captures the dissimilarity between moving northward/southward and moving in other directions – these two move actions do not bring agents near to or far away from enemies.

Although we only show results on two maps, each with one random seed, similar action clusters can be observed on all the scenarios and are robust across different seeds. Therefore, we conclude that our predictive model effectively learns action representations that reflect the effects of actions.

## 4.2 PERFORMANCE AND ABLATION STUDY

For evaluation, all experiments in this section are carried out with 8 different random seeds, and the median performance as well as the 25-75% percentiles are shown. The detailed setting of hyper-parameters is described in Appendix A.

We benchmark our method on all 14 scenarios in order to evaluate its performance across the entire SMAC suite, and the results are shown in Fig. 4. We compare with current state-of-the-art value-based MARL algorithms (QMIX (Rashid et al., 2018), QPLEX (Wang et al., 2020b), Weighted QMIX (Rashid et al., 2020a), and AI-QMIX (Iqbal et al., 2020)), hierarchical MARL method (HSD, Yang et al. (2019)), role-based MARL method (ROMA, Wang et al. (2020c)), and MARL method based on action semantics (ASN, Wang et al. (2020f)). RODE outperforms all the baselines by at least $1/32$ on **9** of all **14** scenarios after $2M$ training steps.

Maps of the SMAC benchmark have been classified as *easy*, *hard*, and *super hard*. Hard and super hard maps are typically hard-exploration tasks. Since RODE is designed to help exploration by decomposing the joint action space, we are especially interested in the performance of our method on these maps. Fig. 3 shows the performance of RODE on all the super hard scenarios and one hard scenario. In Appendix C.1, we further show learning curves on the whole benchmark. We can see on all super hard maps and most hard maps, RODE has the best performance. Moreover, RODE outperforms baselines by the largest margin on the maps that require more exploration: 3s5z_vs_3s6z, corridor, and 6h_vs_8z. In contrast, on 5 easy maps, RODE typically needs more samples to learn a successful strategy. We hypothesize that this is because

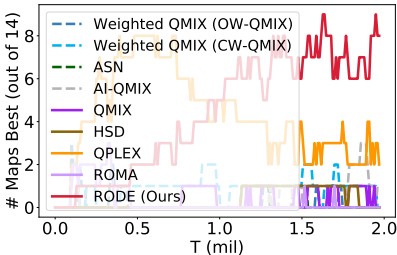

Figure 4: The number of scenarios (out of all 14 scenarios) in which the algorithm's median test win % is the highest by at least $1/32$.

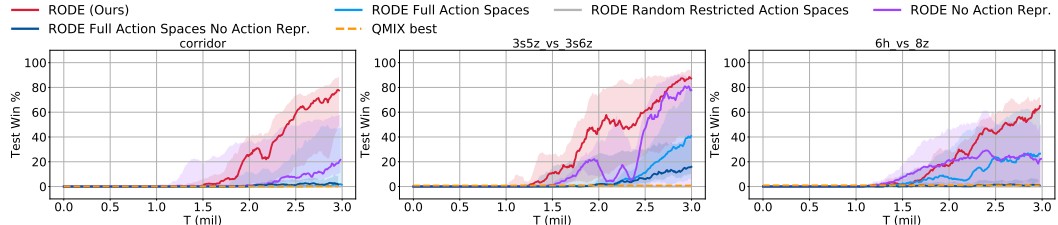

Figure 6: Ablation studies regarding each component of RODE. The best performance that can be achieved by QMIX is shown as horizontal dashed lines.

these maps do not require substantial exploration, and the exploratory benefit brought by restricted action spaces is not obvious.

**Ablations** To understand the superior performance of RODE, we carry out ablation studies to test the contribution of its four main components: (A) Restricted role action spaces; (B) Using action effects to cluster actions; (C) The integration of action representations into role policies and the role selector; (D) The hierarchical learning structure. To test component A and B, when updating role action spaces after $50K$ samples, we let each role action space contains all actions or a random subset of actions, respectively, while keeping other parts in the framework unchanged. For component B, we make sure the union of role action spaces equals the entire action space. To test component C, we can simply use conventional $Q$-networks for learning role policies and the role selector. The testing of component D is a bit different because we cannot ablate it separately while still using different roles. Therefore, we test it by ablating components A and C – using conventional deep $Q$-networks for role policies and the role selector and allowing role policies to choose from all primitive actions.

The results on the three most difficult scenarios from the benchmark are shown in Fig. 6. Generally speaking, performance of RODE using conventional deep $Q$-networks (*RODE No Action Repr.*) is close to that of RODE, and is significantly better than the other two ablations where role action spaces are not restricted. Therefore, using action representations in policy learning can help improve learning but the superior performance of RODE is mostly due to the restriction of the role action spaces. RODE with full role action spaces and conventional $Q$-networks (*RODE Full Action Spaces No Action Repr.*) has similar performance to QMIX, which demonstrates that the hierarchical structure itself does not contribute much to the performance gains. RODE with random restricted role action spaces also cannot outperform QMIX, highlighting the importance of using action effects to decompose joint action spaces.

In summary, by virtue of restricted role action spaces and the integration of action effect information into policies, RODE achieves a new state of the art on the SMAC benchmark. In Appendix B, we provide a case study to better understand why restricting role action spaces can improve performance. We study the effects of the interval between role selections in Appendix D.1.

### 4.3  POLICY TRANSFER

A bonus of basing role policies on action representations is that we can transfer the learned policies to tasks with new actions that have similar effects to old ones. To demonstrate this, given a new task, we first collect some samples ($50k$) and train an action encoder. In this way, we identify the old actions that have similar functionality to the new actions. Then we use the average representation of these old actions as representations for new actions and add new actions to role action spaces to which the similar old actions belong. Since role action spaces are still expected to contain actions with similar effects,

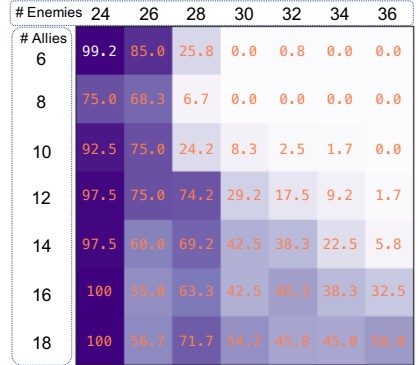

Figure 5: Win rates on unseen maps of the policy learned on `corridor`, where 6 ally Zealots face 24 Zerglings. We do not train policies on new maps.

which can be reflected by old role representations, we do not re-calculate the role representations on

the new task. Moreover, in our framework, it is roles that are interacting with the environment, and agents only need to be assigned with roles. Therefore, our learned policies can also be transferred to tasks with different numbers of agents.

We test the transferability of our method on the SMAC benchmark. To keep the length of observations fixed, we sort allies and enemies by their relative distances to an agent and include the information of nearest $N_a$ allies and $N_e$ enemies in the visible range. In Fig. 5, we show win rates of the policy learned from map `corridor` on different maps without further policy training. In the original task, 6 Zealots face 24 enemy Zerglings. We set $N_a$ to 5 and $N_e$ to 24. We increase the number of agents and enemies, and the number of actions increases as the number of enemies grows. The transferability of RODE is attested by observing that the learned policy can still win 50% of the games on unseen maps with three times the number of agents. More details of this experiment is described in Appendix C.2. Besides rapid transfer, iteratively training the transferred policies on larger tasks may be a promising future direction to scaling MARL to large-scale problems.

## 5    CONCLUSION

Coming up with a set of roles that can effectively decompose the task is a long standing problem preventing role-based learning from realizing scalability. Instead of learning roles from scratch, in this paper, we find that role discovery becomes much easier if we first decompose joint action spaces according to action effects. With a specially designed hierarchical learning framework, we achieve efficient learning over these factored action spaces. We believe that the scalability and transferability provided by our method are crucial in building flexible and general-purpose multi-agent systems.

### ACKNOWLEDGMENTS

We would like to thank the anonymous reviewers for their insightful comments and helpful suggestions. This work is supported in part by Science and Technology Innovation 2030 – "New Generation Artificial Intelligence" Major Project (No. 2018AAA0100904), and a grant from the Institute of Guo Qiang, Tsinghua University.

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

## A ARCHITECTURE, HYPERPARAMETERS, AND INFRASTRUCTURE

In this paper, we use simple network structures for the role selector and role policies. Each role policy is a simple linear network without hidden layers or activation functions, and the role selector is a two-layer feed-forward fully-connected network with a $64$-dimensional hidden layer. Agents share a trajectory encoding network consisting of two layers, a fully-connected layer followed by a GRU layer with a $64$-dimensional hidden state. For all experiments, the length of action representations, $d$, is set to $20$. The outputs of role policies and the outputs of role selector are each fed into their own separate QMIX-style mixing networks (Rashid et al., 2020b) to estimate the global action values. The two mixing networks use the same architecture, containing a 32-dimensional hidden layer with ReLU activation. Parameters of the mixing networks are generated by hypernetworks conditioning on global states. These settings are the same to QMIX (Rashid et al., 2020b). Moreover, it can be easily extended to other mixing mechanisms, such as QPLEX (Wang et al., 2020b), which may further improve the performance of RODE.

For all experiments, the optimization is conducted using RMSprop with a learning rate of $5 \times 10^{-4}$, $\alpha$ of 0.99, and with no momentum or weight decay. For exploration, we use $\epsilon$-greedy with $\epsilon$ annealed linearly from $1.0$ to $0.05$ over $50K$ time steps and kept constant for the rest of the training. For three hard exploration maps—`3s5z_vs_3s6z`, `6h_vs_8z`, and `27m_vs_30m`—we extend the epsilon annealing time to $500K$, for both RODE and all the baselines and ablations. Batches of 32 episodes are sampled from the replay buffer, and the role selector and role policies are trained end-to-end on fully unrolled episodes. All experiments on the SMAC benchmark use the default reward and observation settings of the SMAC benchmark (Samvelyan et al., 2019). Experiments are carried out on NVIDIA GTX 2080 Ti GPU.

We use $k$-means clustering when determining role action spaces. The number of clusters, $k$, is treated as a hyperparameter. Specifically, on maps with homogeneous enemies, we set $k$ to 3, and on maps with heterogeneous enemies, we set $k$ to 5. If the task only involves one enemy, $k$ is set to 2. We can avoid this hyperparameter by using more advanced clustering approaches.

For all baseline algorithms, we use the codes provided by their authors where the hyperparameters have been fine-tuned on the SMAC benchmark.

## B CASE STUDY: ROLE DYNAMICS

In this section, we provide a case study to explore the dynamics of the role selector and give an example to explain why restricting action spaces of roles can improve learning performance.

We carry out our study on the map `corridor` from the SMAC benchmark. We choose this map because it presents a hard-exploration task. An optimal strategy for this scenario requires active state space exploration for the allied Zealots to learn to move to edges or to the choke point on the map, so as to not be surrounded by the enemy army. Previous state-of-the-art algorithms all learn a suboptimal strategy, where agents merely damage the enemy units for rewards, instead of first moving to edges and then attacking.

In Fig. 7, we show the roles assigned by the learned role selector at different timesteps in an episode. The first row shows game snapshots, and the second row presents the corresponding role assignments. Here, the action encoder gives three clusters of actions (see Fig. 2left) – move `eastward` & `northward`, `attack`, and move `westward` & `southward`. These clusters of actions form the action spaces of three roles. The color of circles indicates the role of an ally unit.

We see that all agents select *Role 2* in the beginning (Fig. 7 (a)) to help them move to the edges, which is the good strategy as discussed above. The agents also learn an effective cooperative policy – one of the agents (agent A) attracts most of the enemies (enemy group A) so that its teammates have an advantage when facing fewer enemies (enemy group B). After killing agent A, enemy group A goes through the choke point and arrives their destination. In Fig. 7 (b), the alive agents start to attack this group of enemies. However, the enemies are still very powerful. To win the game, ally agents learn to attract part of enemies and kill them. They achieve this by alternating between *Role 0* (`retract` in this case) and *Role 1* (`attack`), as shown in Fig. 7 (b-d).

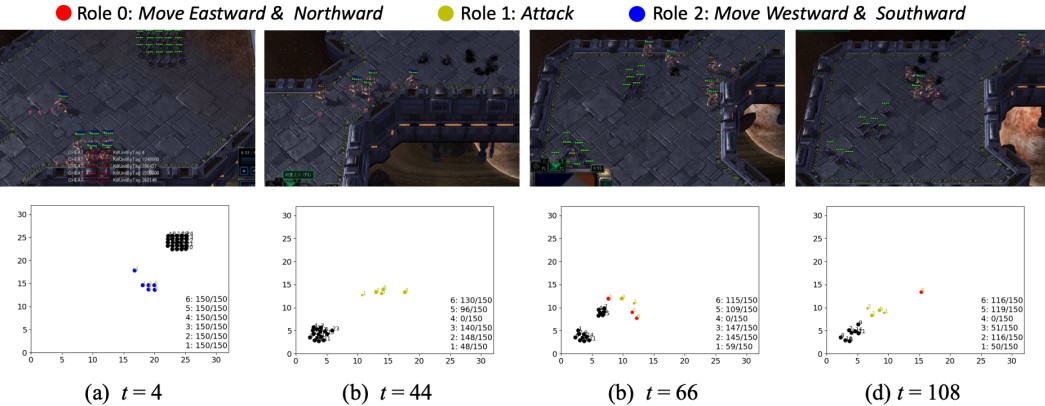

Figure 7: Visualization of roles in one episode on `corridor`. The first row shows game snapshots. In the second row, black circles are enemies and circles in other colors are ally agents, with different colors indicating different assigned roles. The size of circles is proportional to the remaining health points of units. The remaining and maximum health points are also shown in the lower right corner of the figures.

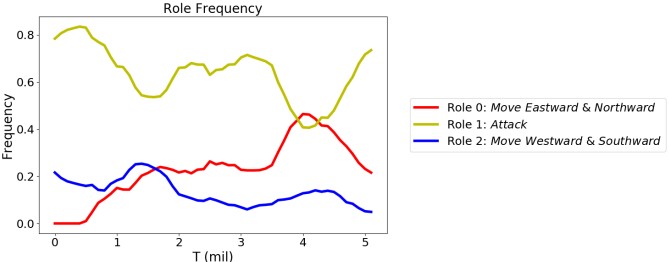

Figure 8: Frequencies of taking each role during training on `corridor`. Correspond to Fig. 7. The frequencies of two roles which help exploration (*Role 0* and *Role 2*) first increase and then decrease.

**Why do restricted action spaces work?** Our effect-based action clustering decomposes the full action spaces into smaller role action spaces, each of which focuses on a smaller subset of actions having similar effects. As each role learns on a smaller action space which has similar actions, it enables agents to efficiently explore and results in each role being able to learn different strategies, like spread and retracting. Moreover, exploration in the much smaller role space makes the coordination of these sub-strategies easily.

Exploring in the hierarchically decomposed action spaces also provides a bias in exploration space. For example, on `corridor`, *Role 0* and *Role 2* motivate agents to explore the state space in a certain direction which help them learn several important strategies on this map: 1) Agents first move to the edges (*Role 0*) to avoid being surrounded by the enemies (Fig. 7 (a)), and 2) Agents alternate between attacking (*Role 1*) and retracting (*Role 2*) to attract and kill part of the enemies, gaining an advantage in numbers (Fig. 7 (b-d)). The fact that *Role 0* and *Role 2* help exploration is also supported by Fig. 8 – their frequencies first improve as the agents explore more, and then decrease when cooperation strategies gradually converge.

On other maps, restricted role action spaces may help agents solve the task in other ways. For example, on the map `3s5z_vs_3s6z`, agents learn more efficiently because the action encoder decomposes the task into two sub-tasks, one focusing on attacking Zealots and another one focusing on attacking Stalkers. Learning policies for these restricted sub-problems significantly reduces the learning complexity. In summary, action effects provide effective information to decompose joint action spaces. Clustering actions according to their effects makes role discovery much easier, and learning cooperative strategies in smaller action spaces makes learning more tractable.

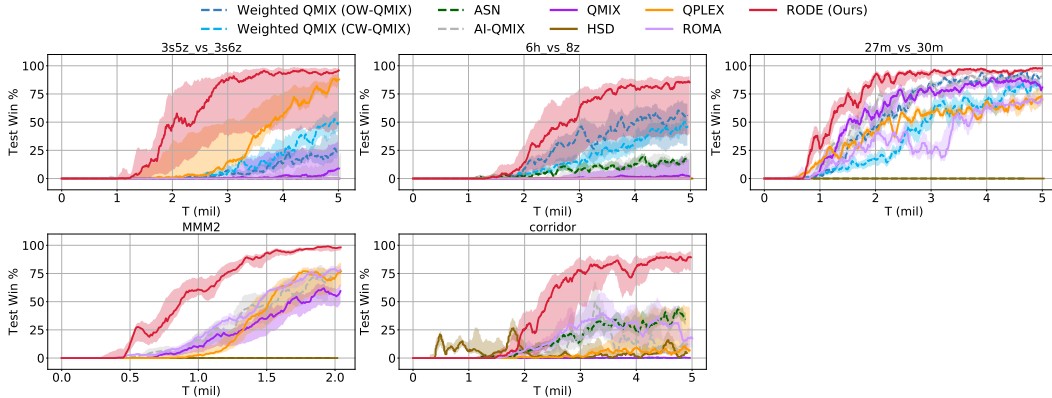

Figure 9: Comparisons between RODE and baselines on all **super hard** maps.

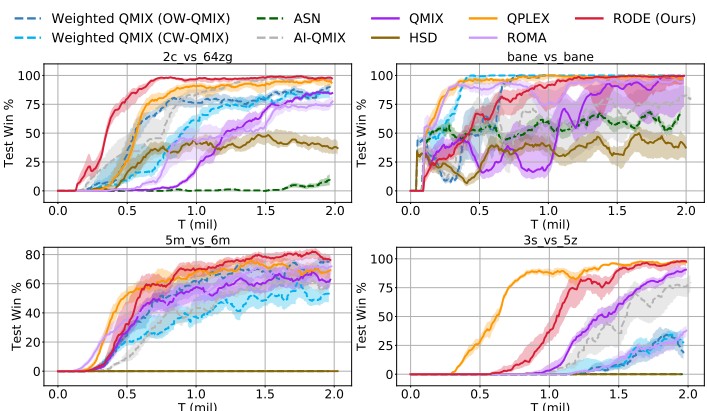

Figure 10: Comparisons between RODE and baselines on all **hard** maps.

## C  EXPERIMENTAL DETAILS

In this section, we provide more experimental results supplementary to those presented in Sec. 4.2. We also discuss the details of experimental settings of rapid policy transfer and analyze the results.

### C.1  BENCHMARKING ON STARCRAFT II MICROMANAGEMENT TASKS

In Sec. 4.2, we show the overall performance of RODE on the SMAC benchmark. In this section, we further show the performance of our method on all maps. All experiments in this section are carried out with 8 random seeds. Median win rate, together with 25-75% percentiles, is shown.

The SMAC benchmark (Samvelyan et al., 2019) contains 14 maps that have been classified as easy, hard, and super hard. In Fig. 9, we compare the performance of RODE with baseline algorithms on all super hard maps. We can see that RODE outperforms all the baselines by a large margin, especially on those requiring the most exploration: 3s5z_vs_3s6z, corridor, and 6h_vs_8z. These results demonstrate that RODE can help exploration and solve complex tasks, in line with our expectations of it.

On hard maps, RODE still has superior performance compared to baselines, as shown in Fig. 10. However, on easy maps (Fig. 11), RODE tends to use more samples to achieve similar performance to baselines. We hypothesize that this is because easy maps do not require substantial exploration – ally agents can win by merely engaging in fight and damaging the enemy units for reward. Under these circumstances, RODE still explores a lot, which prevents it from performing better.

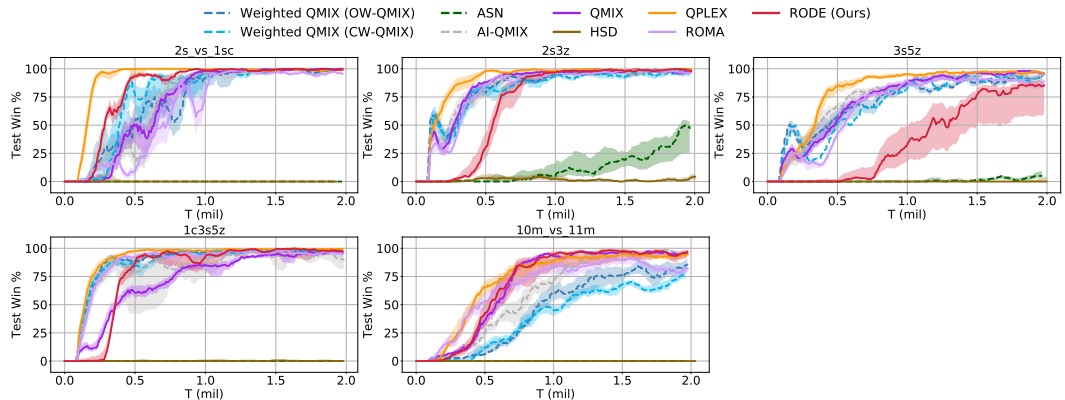

Figure 11: Comparisons between RODE and baselines on all **easy** maps.

In summary, RODE establishes a new state of the art on the SMAC benchmark by outperforming all baselines on 9 out of 14 scenarios. Notably, its superiority is particularly obvious on all of the hard and super hard maps.

## C.2 POLICY TRANSFER

In this section, we describe the setting of the policy transfer experiment introduced in Sec. 4.3.

Policy transfer experiments are based on the assumption that new actions have similar functionalities to some old actions. We train an action encoder on the new task to cluster the actions. In this way, we can distinguish old actions $a_{i_1}, \ldots, a_{i_m}$ that have similar effects to a new action $a_{n_i}$. We then use $\frac{1}{m} \sum_{j=1}^{m} z_{a_{i_j}}$ as the representation for $a_{n_i}$, where $z_{a_{i_j}}$ is the representation for $a_{i_j}$ on the original task, and we add $a_{n_i}$ to role action spaces to which $a_{i_1}, \ldots, a_{i_k}$ belong. For each new action, we repeat these operations. The number of roles does not change during this process. This process enables the role selector and role policies learned in original task to be rapidly transferred to a similar task.

To test our idea, we train RODE on the map `corridor`, where 6 ally Zealots face 24 enemy Zerglings, for 5 million timesteps and rapidly transfer the learned policy to similar maps by increasing the number of agents and enemies. The number of actions also increases because a new `attack` action is added for every new enemy unit. The results are shown in Fig. 5.

## D ROLE INTERVAL AND RECURRENT ROLE SELECTOR

### D.1 ROLE INTERVAL

In our framework, the role selector assigns roles to each agent every $c$ timesteps. We call $c$ the role interval. The role interval decides how frequently the action spaces change and may have a critical influence on the performance. To get a better sense of this influence, we change the role interval while keeping other parts unchanged and test RODE on several environments. In Fig. 12, we show the results on 2 easy maps (`10m_vs_11m` and `1c3s5z`), 1 hard map (`3s_vs_5z`), and 1 super hard map (`6h_vs_8z`). Generally speaking, the role interval has a significant influence on performance, but 5 or 7 can typically generate satisfactory results. In this paper, we use a role interval of 5 on most of the maps.

### D.2 RECURRENT ROLE SELECTOR AND FULLY-CONNECTED ROLE SELECTOR

The role selector is supposed to learn a decomposition of observation space. However, we base this component on local action-observation history. In this section, we discuss the influence of using GRUs in the role selector. In Fig. 13, we compare RODE to *RODE with a fully-connected role selector* conditioned on local observations. As expected, a recurrent role selector performs better

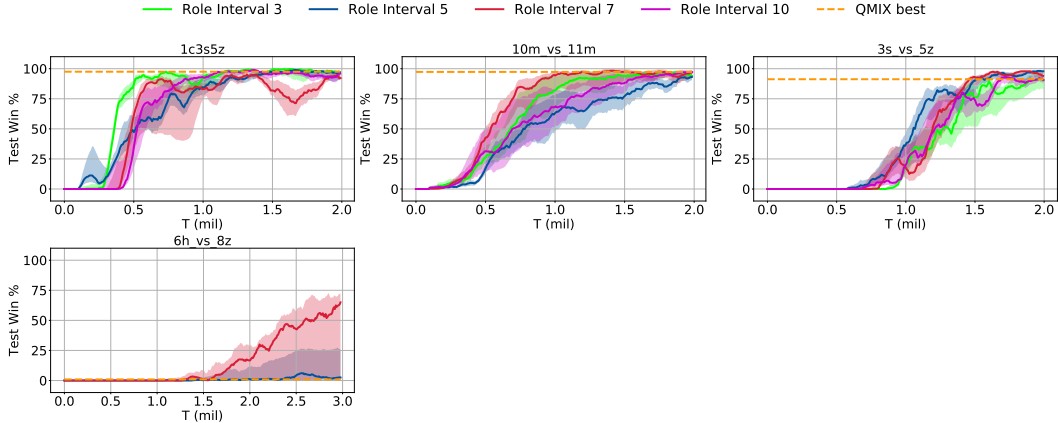

Figure 12: Influence of the role interval (the number of timesteps between two consecutive role selections).

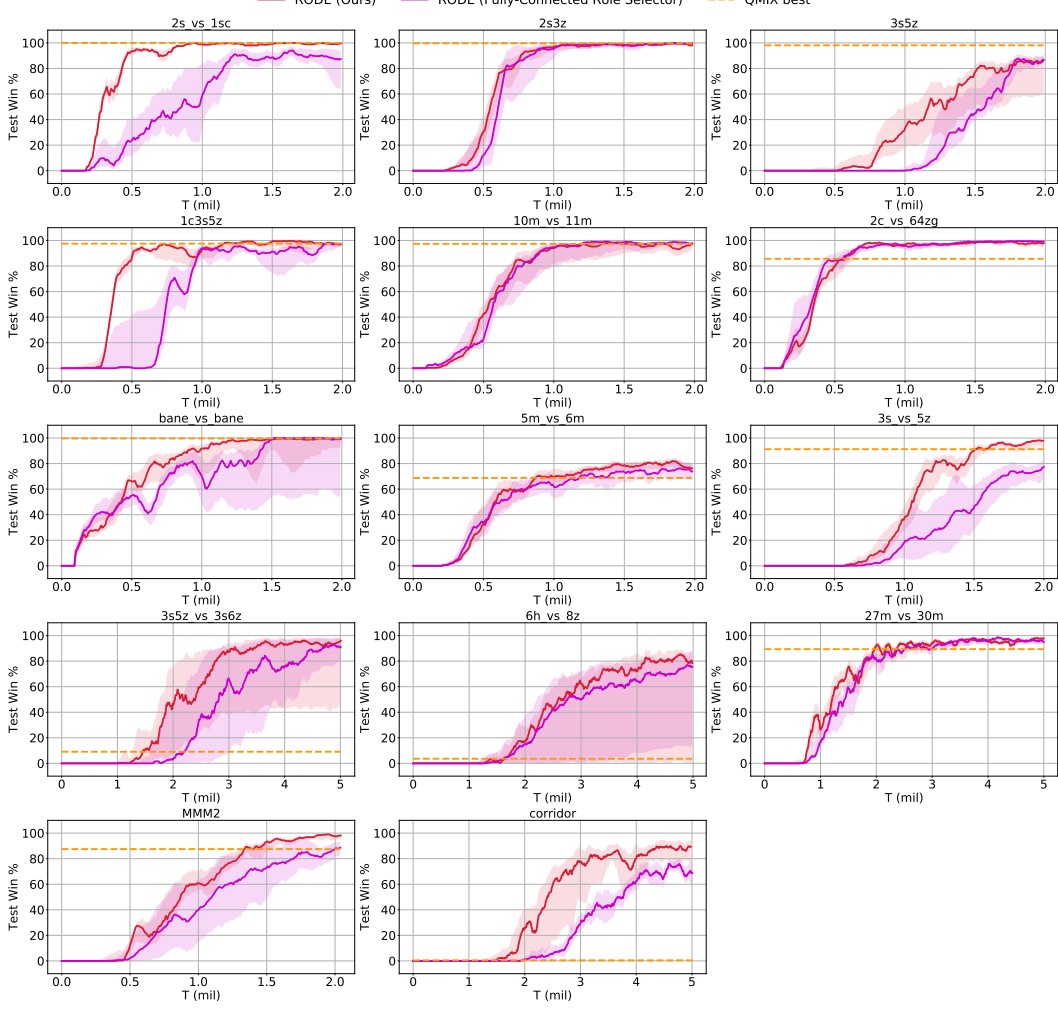

Figure 13: Comparisons between RODE with a recurrent role selector (RODE) and a fully-connected role selector conditioned on local observations (RODE (Fully-Connected Role Selector)).

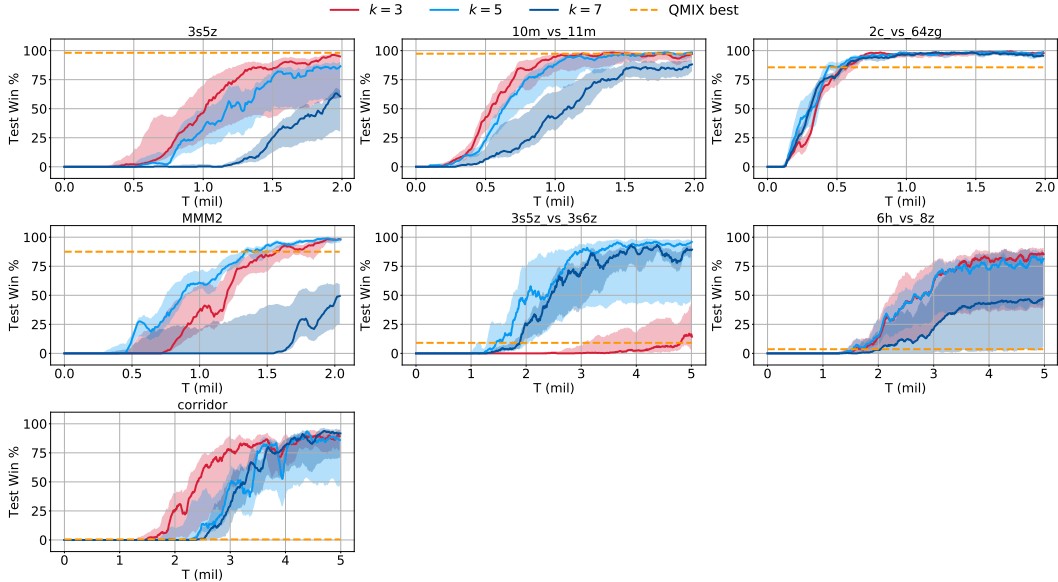

Figure 14: RODE with different $k$ values.

than a selector based on local observations. Even though this conclusion is not surprising, we are interested in those environments where a recurrent role selector can not perform better. For map 10m_vs_11m, together with the results presented in Fig. 12, we see that RODE with a role interval of 7 has similar performance to a fully-connected role selector, but underperforms when the role interval is set to other values. This indicates that the role interval has a sophisticated interaction with exploration, optimization, and learning stability. We leave the study of this as future work. In the previous section, we give more examples showing the influence of the role interval.

## E  CLUSTERING ALGORITHMS

In our framework, we use $k$-means to cluster action representations to determine the role action spaces. $K$-means algorithm requires predefining the number of clusters $k$. In this section, we first investigate the influence of $k$ values on the performance and then discuss how to determine the number of clusters with minimum prior knowledge.

### E.1  INFLUENCE OF K VALUES

We test the influence of $k$ values on 7 maps, including 2 easy maps (3s5z and 10m_vs_11m), 2 hard maps (2c_vs_64zg and MMM2), and 3 super hard maps (3s5z_vs_3s6z, 6h_vs_8z, and corridor). In Fig. 14, we show the performance of RODE with $k$-values of 3, 5, and 7. Generally speaking, a $k$-value of 3 or 5 can work well across these maps. In practice, we set $k = 3$ on maps with homogeneous enemies and $k = 5$ on maps with heterogeneous enemies. For outliers, we add them to all other clusters to avoid action sets consisting of only one action, which we find can largely stabilize training across the benchmark.

To further understand how different values of $k$ affect the learning performance, we take one super hard map, 6h_vs_8z, as an example. When $k$=3 or 5, the clustering method (after adding outliers to all other clusters) gives three role action spaces – Move Eastward & Northward, Attack, and Move Westward & Southward. Similar to the case of corridor as discussed in Appendix B, these roles can improve exploration and thus boost the performance. However, when $k$=7, all move actions become outliers. The result is that most role action spaces become the original full action space. Such a decomposition over action space does not reduce the search space, and thus the performance is close to that of QMIX.

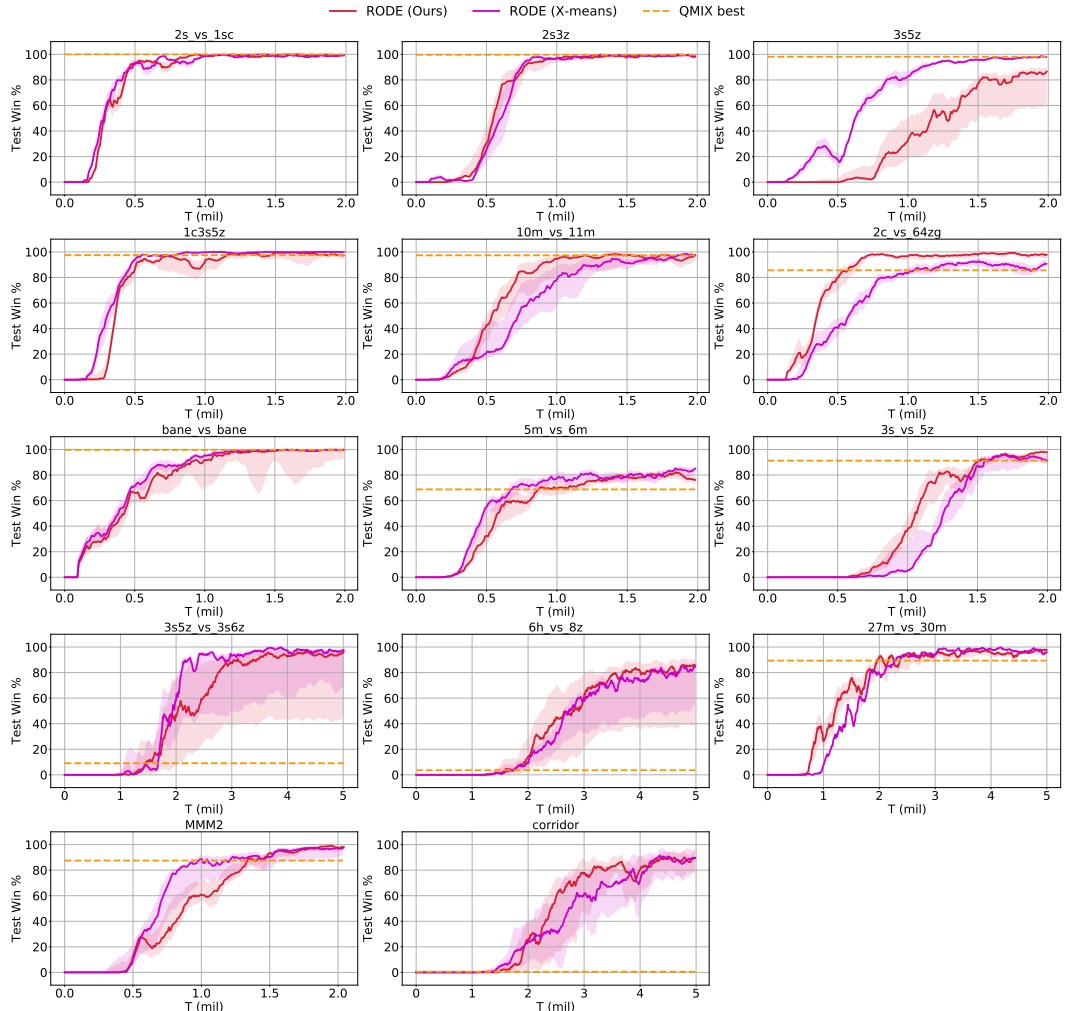

Figure 15: Comparisons between RODE using X-means clustering and RODE using $k$-means clustering, where $k = 3$ for maps with homogeneous enemies and $k = 5$ for maps with heterogeneous enemies. X-means does not require predefining the number of clusters.

To avoid such complicated interactions between the selection of $k$ values and the way of dealing with outliers, we propose to use $X$-means when clustering action representations, as we discuss in detail in the next section.

## E.2 OTHER CLUSTERING ALGORITHMS

In this section, we investigate how to avoid predefining $k$ values.

X-means (Pelleg et al., 2000) improves $k$-means by automatically determining the number of clusters. We use *minimum noiseless description length* as the splitting criterion for X-means and test RODE using X-means on all the 14 maps. The results are shown in Fig. 15. We see that the results are comparable to $k$-means with predefined $k$ values. We thus recommend X-means when facing a new task, while for tasks that users know the number of roles, $k$-means can be more accurate.

Additionally, we can use density-based clustering algorithms to avoid the predefinition of $k$, like DBSCAN (Ester et al., 1996). In practice, we find that DBSCAN can generate similar role action spaces to X-means. Since the clustering algorithm is used only to split the action spaces, RODE using DBSCAN clustering is expected to have similar learning performance to X-means.

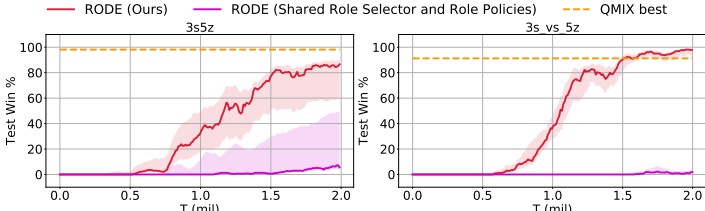

Figure 16: Comparisons between RODE and RODE with shared role policies and role selector. The separation of the role selector and role policies is important to make the average of action representations well represent the roles.

## F    ROLE REPRESENTATIONS

In our framework, we use the average representation of actions in the role action space to represent a role. One concern is that whether such representations can well represent the roles, given that both role policies and the role selector are based on action representations. We find that the separation of Q networks for the role selector and role policies (including the GRUs for processing local observations and the network $f_\beta$ (or $f_{\rho_j}$)) is important to ensure such role representations have sufficient expressivity. If we share the GRUs and $f_\beta$ (or $f_{\rho_j}$), RODE struggles even on easy maps (Fig. 16).

## G    RELATED WORKS

**Role-Based Learning**    Our approach is an attempt to integrate roles into deep multi-agent reinforcement learning. Many natural systems feature emergent roles, such as ants (Gordon, 1996), bees (Jeanson et al., 2005), and humans (Butler, 2012). In these systems, roles are closely related to the division of labor and is critical to labor efficiency. These benefits inspired multi-agent system designers, who try to reduce the design complexity by decomposing the task and specializing agents with the same role to certain sub-tasks (Wooldridge et al., 2000; Omicini, 2000; Padgham & Winikoff, 2002; Pavón & Gómez-Sanz, 2003; Cossentino et al., 2005; Zhu & Zhou, 2008; Spanoudakis & Moraitis, 2010; DeLoach & Garcia-Ojeda, 2010; Bonjean et al., 2014). However, roles and the associated responsibilities (or subtask-specific rewards (Sun et al., 2020)) are predefined using prior knowledge in these systems (Lhaksmana et al., 2018). Although pre-definition can be efficient in tasks with a clear structure, such as software engineering (Bresciani et al., 2004), it hurts generalization and requires prior knowledge that may not be available in practice. To solve this problem, Wilson et al. (2010) use Bayesian inference to learn a set of roles and Wang et al. (2020c) design a specialization objective to encourage the emergence of roles. However, these methods search the optimal task decomposition in the full state-action space, resulting in inefficient learning in hard-exploration tasks. RODE avoids this shortcoming by first decomposing joint action spaces according to action effects, which makes role discovery much easier.

**Multi-Agent Reinforcement Learning (MARL)**    Many challenging tasks (Vinyals et al., 2019; Berner et al., 2019; Samvelyan et al., 2019; Jaderberg et al., 2019) can be modelled as multi-agent learning problems. Many interesting real-world phenomena, including the emergence of tool usage (Baker et al., 2020), communication (Foerster et al., 2016; Sukhbaatar et al., 2016; Lazaridou et al., 2017; Das et al., 2019), social influence (Jaques et al., 2019), and inequity aversion (Hughes et al., 2018) can also be better explained from a MARL perspective.

Learning control policies for multi-agent systems remains a challenge. Centralized learning on joint action space can avoid non-stationarity during learning and may better coordinate the learning process of individual agents. However, these methods typically can not scale well to large-scale problems, as the joint action space grows exponentially with the number of agents. The coordination graph (Guestrin et al., 2002b;a; Böhmer et al., 2020) is a promising approach to achieve scalable centralized learning. It reduces the learning complexity by exploiting coordination independencies between agents. Zhang & Lesser (2011); Kok & Vlassis (2006) also utilize the sparsity and locality of coordination. These methods require the dependencies between agents to be pre-supplied. Value function decomposition methods avoid using such prior knowledge by directly learning centralized

but factored global $Q$-functions. They implicitly represent the coordination dependencies among agents by the decomposable structure (Sunehag et al., 2018; Rashid et al., 2018; Son et al., 2019; Wang et al., 2020e;b; Rashid et al., 2020a; Iqbal et al., 2020).

Multi-agent policy gradient algorithms enjoy stable theoretical convergence properties compared to value-based methods (Gupta et al., 2017; Wang et al., 2020a) and hold the promise to extend MARL to continuous control problems. COMA (Foerster et al., 2018) and MADDPG (Lowe et al., 2017) propose the paradigm of centralized critic with decentralized actors (CCDA). PR2 (Wen et al., 2019) and MAAC (Iqbal & Sha, 2019) extend the CCDA paradigm by introducing the mechanism of recursive reasoning and attention, respectively. DOP (Wang et al., 2020g) solves the centralized-decentralized mismatch problem in the CCDA paradigm and enables deterministic multi-agent policy gradient methods to use off-policy data for learning. Another line of research focuses on fully decentralized actor-critic learning (Macua et al., 2017; Zhang et al., 2018; Yang et al., 2018; Cassano et al., 2018; Suttle et al., 2019; Zhang & Zavlanos, 2019).

