# OpenReview forum: "RODE: Learning Roles to Decompose Multi-Agent Tasks"
_ICLR.cc/2021/Conference — ICLR 2021 Poster_

### Official Review · AnonReviewer1 · 2020-10-28

**Rating:** 6
**Confidence:** 4

**Review:**

This paper proposes a new method called RODE, to learn roles for multiagent systems to improve the learning efficiency of MARL. Instead of exploring in the full joint action space, RODE first decomposes the action space into K clusters based on the different influences of each action on the environment and other agents. Then RODE trains a role selector and role policies conditioned on each agent’s action-observation history. Experiments show its superior performance compared with SOTA MARL algorithms.

********
This paper is highly related to the subject of ICLR and is well written. But I have some specific confusions listed below, waiting for the authors' reply.

The authors mentioned that RODE differs from previous role-based methods that require prior knowledge to learn roles. However, the action space decomposition also requires human knowledge, e.g., would a wrong number of k clusters hinder the final performance?

Given the local observation and all agent’s actions, RODE first learns the action representations by minimizing two prediction errors: the next local observation, and team reward. Then the role representation is calculated by averaging representations of actions in each cluster. What I am concerned is that whether this average representation can well represent the roles, which may further impact the estimation of role policies, since the action representations are used to generate the q-values of both role selector and role policies.


Since in SMAC, the attack action must be given an enemy index, so the number of actions increases with the increase of the number of enemies. Transfer of RODE needs to manually add new representations to each cluster and then re-calculate the role representations, In this way, the role representations change. Does this influence the decisions of role selector and role policies? Furthermore, these two components cannot be trained on new maps, because the input and output of the mixing networks are different with a different number of agents.

Some recent MARL algorithms to improve QMIX, such as AI-QMIX[1], Weighted-QMIX[2] should be discussed or compared.

I agree with the authors that the joint action space can be decomposed or classified based on the action effect or action semantics. ASN [2] firstly investigates the influence of actions on other agents, which they call the action semantics. I think these two works are inspired by the same property in MASs to improve multiagent coordination. Moreover, these two works both use SMAC as benchmarks. In this way, I’d like to see the comparison of ASN and RODE to further validate its superior performance.

[1] AI-QMIX: Attention and Imagination for Dynamic Multi-Agent Reinforcement Learning. arXiv preprint arXiv:2006.04222.

[2] Weighted QMIX: Expanding Monotonic Value Function Factorisation. NeurIPS 2020.

[3] Action Semantics Network: Considering the Effects of Actions in Multiagent Systems. ICLR. 2020.

---

> ### Author Response · Authors · 2020-11-18
> **Thanks for the insightful review. We provide new experimental results regrading clustering, role representations, and related works. We also provide clarifications for your other concerns.**
>
> We thank the reviewer for the insightful review.
>
> **Q1**: "The action space decomposition also requires human knowledge, e.g., would a wrong number of $k$ clusters hinder the final performance?"
>
> **A1**: We investigate the effect of $k$ by running RODE with different $k$ values on 7 maps (2 easy maps, 2 hard maps, and 3 super hard maps). Results are shown in Appendix E.1 (Fig. 14 on page 21) of the updated version. Across these maps, we find a $k$ value of 3 or 5 generally works well, but a larger $k$ value hurts performance (detailed reasons and examples are provided in Appendix E.1). To avoid selecting wrong $k$ values for new tasks, we propose to use X-means clustering to automatically determine the number of clusters. In Appendix E.2, we compare RODE using X-means with $k$-means on all the 14 maps and find that the performance of RODE using X-means is similar to $k$-means with predefined $k$ values. These results show that we can avoid requiring human knowledge by decomposing the action space using X-means clustering.
>
> **Q2**: "What I am concerned about is that whether this average representation can well represent the roles, which may further impact the estimation of role policies, since the action representations are used to generate the $Q$-values of both role selector and role policies."
>
> **A2**: In Appendix F of the updated paper, we show that if the role selector and role policies are shared (including GRUs for processing local observations and the network $f_\beta$ (or $f_{\rho_j}$)), the average representations cannot represent the roles well (Fig. 16 on page 23). By contrast, in RODE the role selector and role policies are separate, and role/action representations are fixed during most of the training stage (fixed after 50$k$ steps, and the training typically lasts for 2$M$ steps). Therefore, role representations and action representations work independently when estimating $Q$-values at different levels. In this way, the average representation can represent the roles well, as supported by our empirical results.
>
> **Q3**: Would the re-calculating role representations influence the decisions of role selector and role policies?
>
> **A3**: As the reviewer stated, the number of actions increases on new maps and policies cannot be trained on new maps. The transfer experiments of RODE consider these issues. We did not re-calculate role representations on new maps.
>
> For a new task, we first train an action encoder to identify the old actions that have similar effects to each new action. We then add new actions to the cluster to which these similar old actions belong. Since the role action spaces are still expected to contain actions with similar effects after addition, we use old role representations for the role selector on the new task. As for the action representations used in role policies, we use old representations for old actions and the average old representation of similar old actions for representing new actions. In this way, the policy can be transferred to new maps with actions of similar effects. We have updated the corresponding descriptions in the first paragraph of Sec. 4.3.
>
> **Q4**: Discussions of and comparisons with related works.
>
> **A4**: We compare RODE against AI-QMIX, weighted QMIX, and ASN on all 14 SMAC maps. Results are shown in Fig. 3, 4 and Appendix C (Fig. 9, 10, and 11 on page 18-19) of the updated version. RODE outperforms these algorithms by a large margin on all super hard maps and most hard maps. Moreover, citations to or discussions of these papers are added in Sec. 3 or Appendix G.
>
> The code and hyper-parameter settings of AI-QMIX for running standard SMAC tasks are provided by its authors. For weighted QMIX, we test both variants, OW-QMIX and CW-QMIX, and the hyper-parameter settings are also provided by its authors. For ASN, we use the code and hyper-parameter settings available at GitHub.

---

### Official Review · AnonReviewer2 · 2020-10-28
**Official Blind Review**

**Rating:** 7
**Confidence:** 4

**Review:**

This paper introduces a bi-level hierarchical framework for achieving scalable multi-agent learning. In this framework, the high-level policy (role selector) coordinates role assignments in a smaller role space and at a lower temporal resolution. And the low-level policies (role policies) explore strategies in reduced primitive action-observation spaces. In this way, the complex multi-agent problem is decomposed into multiple sub-problems, which is easy to learn. The authors conduct the experiments in the StarCraft II micromanagement benchmark, compared with the state-of-the-art MARL methods.

Strength:
- The paper is well-written and easy-to-follow. The authors show the demo videos for a better demonstrating of the learned behavior.
- The introduced idea is original and interesting. I can see that the bi-level multi-agent coordination framework is of great potential in solving various multi-agent tasks.
- It is nice to demonstrate the generalization of the learned policies on unseen maps.
- Provide the code in the Supplementary Material.

Questions:
- The action clustering. How to choose the roles number k for clustering? According to prior knowledge? Can you explain the effect of different k numbers on the performance?
- Is the policies for different levels trained simultaneously?

---

> ### Author Response · Authors · 2020-11-18
> **Thanks for the thoughtful comments. We provide experimental results discussing the effects of different $k$ numbers and how to automatically choose the number of clusters. We also discuss the training of different levels.**
>
> We thank the reviewer for the thoughtful comments.
>
> **Q1**: "How to determine $k$? The effects of different $k$ on the performance?"
>
> **A1**: We investigate the effect of different $k$ on the performance by setting $k$ to be 3, 5, 7 and test RODE on 7 maps (2 easy maps, 2 hard maps, and 3 super hard maps). Results are shown in Appendix E.1 (Fig. 14 on page 21) of the updated version. Generally speaking, a $k$ value of 3 or 5 works well across the tasks. However, a larger $k$ value hurts the performance (detailed reasons and examples can be found in Appendix E.1). To automatically determine a suitable number of clusters, we propose to use X-means clustering. We show the performance of RODE using X-means clustering on all 14 SMAC maps in Appendix E.2 (Fig. 15 on page 22) of the updated paper. We see that X-means achieves similar performance to $k$-means with predefined $k$ values.
>
> **Q2**: "Are the policies for different levels trained simultaneously?"
>
> **A2**: Yes, they are simultaneously trained. For training, we sample 32 episodes from the replay buffer. Then the fully unrolled episodes are used to update both the role selector and role policies, simultaneously.

---

### Official Review · AnonReviewer3 · 2020-10-29
**A paper with a clear contribution**

**Rating:** 8
**Confidence:** 3

**Review:**

This paper describes a role-based learning model for the DEC-POMDPs. The main contribution lies in the efficient discovery of roles from the joint action spaces and then learning a bi-level role assignment for each achievement. This is achieved in two steps. First, the joint action space is clustered into different role action spaces that reduce the action search space for each role. Second, a bi-level role assignment technique is used to learn action and roles for each agent. The technique is tested on StarCraft II micromanagement environments.

For the action space reduction, the model learns action representations that can reflect the effects of actions on the environment and other agents. To this end, a deep learning model is created which predicts the effects of joint actions on the induced rewards and change in the effects. Actions generating similar effects are cluster together using K-means and are called roles action spaces. This restricts the joint action search spaces for each role. The role selector is now used to learn a bi-level hierarchical assignment to map the action-observation history of each agent. At the top-level the agents are mapped to their corresponding to roles based on a Q-value function of each role conditioned on action-observation history and at a lower-level similar Q-value function is used to find the agent’s action. To avoid too many concurrent selections of a single role and action by multiple agents, a global Q-value is learned from individual Q-values to ensure overall coordination between the agents. This is inspired by QMIX, previous work on multi-agent learning.

Positives:
1. The idea of reducing the search space by effect-based clustering appears interesting and novel.
2. The technique leads to good exploration and performance in hard and super maps.
3. The paper is well-written, and the technique is extensively tested on all the maps with useful ablations

Minor issues:
1. Some comments/reasoning related to outlier roles and action spaces would have been helpful
2. Do changes in the clustering algorithm leads to a significant difference in performance or role assignment?
3. Compared to the previous approaches, the RODE-algorithm learns slower in most of the easier maps.

Update - Thank you for the response and updates to the paper

---

> ### Author Response · Authors · 2020-11-18
> **We thank the reviewer for the inspiring comments. We provide experimental results and analyses regarding outlier action spaces (or roles) and different clustering algorithms. We also discuss the performance of RODE on easier maps.**
>
> We thank the reviewer for the inspiring comments.
>
> **Q1**: "Comments/reasoning related to outlier roles and action spaces."
>
> **A1**: We add outlier action spaces (those consisting of only one action) to all other clusters to prevent role action spaces from consisting of only one action. We find this can help stabilize training. However, this mechanism also has some drawbacks. For example, when $k$-means is carried out with a too large value of $k$, many outliers will occur. After adding them to all other role action spaces, most roles will have a role action space that is similar to the full action space. Consequently, RODE cannot discover roles that can effectively reduce the search space. In Appendix E.1 of the updated paper, we use detailed examples to show the interaction between $k$ values and this way of dealing with outliers. To avoid this problem, in Appendix E.2, we show that using X-means clustering can automatically determine the number of clusters and performs comparably to $k$-means clustering with predefined $k$ values.
>
> **Q2**: "Do changes in the clustering algorithm lead to a significant difference in performance or role assignment?"
>
> **A2**: In Appendix E.2 of the updated paper, we compare RODE (using $k$-means clustering) against RODE using X-means clustering on all 14 SMAC maps. The results show that they achieve similar performance (Fig. 15 on page 22). Additionally, we test DBSCAN clustering and find that the action space is decomposed similarly to that of X-means.
>
> **Q3**: "Compared to the previous approaches, the RODE-algorithm learns slower in most of the easier maps."
>
> **A3**: One reason why RODE can boost performance is the improved exploration, as discussed in Appendix B. An accompanying problem is that RODE explores a lot even in maps that do not require much exploration, like in the case of most easier maps. We think that an adaptive exploration-exploitation tradeoff mechanism is a promising research direction for deep role-based MARL.

---

### Decision · Program_Chairs · 2021-01-07
**Final Decision**

**Decision:**

Accept (Poster)

**Comment:**

The paper proposes a two-level hierarchical algorithm for efficient and scalable multi-agent learning where the high-level policy decides a reduced space for low-level to explore in. All the reviewers liked the premise and the experimental evaluation. Reviewers had some clarification questions which were answered in the authors' rebuttal. After discussing the rebuttal, AC as well as reviewers believe that the paper provides insights that will be useful for the multi-agent learning community and recommend acceptance.